astrophysics/stars/relativity

white dwarf, Chandrasekhar limit, generalized uncertainty principle, dynamical instability, gravitational collapse

**Author for correspondence:**
Malay K. Nandy
e-mail: mknandy@iitg.ac.in

# Existence of Chandrasekhar's limit in generalized uncertainty white dwarfs

## Arun Mathew and Malay K. Nandy

Department of Physics, Indian Institute of Technology Guwahati, Guwahati 781039, India

AM, 0000-0001-9896-4243; MKN, 0000-0002-2640-0497

The existence of Chandrasekhar's limit has played various decisive roles in astronomical observations for many decades. However, various recent theoretical investigations suggest that gravitational collapse of white dwarfs is withheld for arbitrarily high masses beyond Chandrasekhar's limit if the equation of state incorporates the effect of quantum gravity via the generalized uncertainty principle. There have been a few attempts to restore the Chandrasekhar limit but they are found to be inadequate. In this paper, we rigorously resolve this problem by analysing the dynamical instability in general relativity. We confirm the existence of Chandrasekhar's limit as well as stable mass–radius curves that behave consistently with astronomical observations. Moreover, this stability analysis suggests gravitational collapse beyond the Chandrasekhar limit signifying the possibility of compact objects denser than white dwarfs.

## 1. Introduction

Chandrasekhar's limit has played a crucial role in numerous astronomical findings for many decades. It is well known that the existence of Chandrasekhar's limit results in Type Ia supernovae (SN Ia) from the explosion of carbon–oxygen (degenerate core) white dwarfs due to accretion from a companion star. Such supernovae have well-defined light-curves with nearly the same peak brightness and their maximum brightnesses have a definite correlation with their light curve decline rates. This property makes them standard candles in astronomy, facilitating measurements on high-redshift Type Ia supernovae, and revealing the accelerated expansion of the Universe [1,2]. Importantly, this ground-breaking finding is based on the existence of Chandrasekhar's limit.

However, it has recently been argued that the generalized uncertainty principle (GUP) removes the Chandrasekhar limit [3–6]. This is due to the fact that the inclusion of GUP,

$$\Delta x \Delta p_x \geq \frac{\hbar}{2}\{1 + \beta(\Delta p_x)^2\}, \tag{1.1}$$

via the equation of state gives white dwarfs of excessively high masses, irrespective of the smallness of the parameter $\beta$. In other words, the mass is no longer bound from above, so that

$$M_{\mathrm{GUP}} = 12\sqrt{2}\left(\frac{\hbar c}{G}\right)^{3/2}\frac{1}{(\mu_e m_u)^2}\ (\beta p_{\mathrm{Fc}}^2)^{3/4}, \tag{1.2}$$

in the high momentum limit. This implies that the GUP-enhanced equation of state prevents gravitational collapse and halts the formation of compact astrophysical objects denser than white dwarfs. This prediction contradicts astronomical observations that confirm the existence of pulsars [7–9] and black holes [10–12]. Moreover, it has been observed that the masses of white dwarfs fall well within the Chandrasekhar limit [13–15],

$$M_{\mathrm{Ch}} = 2.0182\frac{\sqrt{3\pi}}{2}\left(\frac{\hbar c}{G}\right)^{3/2}\frac{1}{(\mu_e m_u)^2} \approx \frac{5.76}{\mu_e^2}M_{\odot}, \tag{1.3}$$

apart from the super-Chandrasekhar white dwarfs, that may well be double-degenerate mergers [16–18].

A solution to the problem was proposed by imposing a cutoff in the Fermi momentum at the neutronization threshold [19]. Since the process of neutronization is not built into the dynamical equations, and it is imposed *by hand*, this solution is not a dynamical consequence of the theory. A more satisfying solution ought to be based on a theory where a collapse happens as a dynamical consequence of the underlying equations of the theory.

It is important to note that excessive mass of white dwarfs results when the GUP parameter $\beta$ is considered to be positive. Theories of quantum gravity suggest a grainy structure of the space–time which naturally implies a minimum uncertainty in position measurement [20–24]. The minimum uncertainty in length in one of the GUP scenarios is given by $\Delta x_{\mathrm{min}} = \hbar\sqrt{\beta}\sqrt{1 + \beta\langle\mathbf{p}\rangle^2}$ as shown by Kempf [25]. Although this implies that $\beta$ is a positive quantity due to $\Delta x_{\mathrm{min}}$ being real valued, there have been various other scenarios [26–28] which suggest that $\beta$ may also be a negative quantity.

For example, a comparison between GUP corrected black hole temperature with that following from a deformed Schwarzschild metric suggested a negative value of the GUP parameter $\beta$ [26,27]. The same suggestion was made [28] by a comparison between the non-commutative space–time correction to the black hole temperature with the GUP corrected black hole temperature.

However, the sign ambiguity of the GUP parameter $\beta$ is still an unresolved problem. For example, on the basis of horizon quantum mechanics [29], it was suggested that $\beta$ should be negative. On the other hand a corpuscular scenario of gravity, where a black hole is pictured as a Bose–Einstein condensate of gravitons, led to a positive value of $\beta$ [30]. It may also be noted that a lattice model with Planckian lattice constant resulted in a negative sign for the GUP parameter $\beta$ [31]. This is in contrast with a stringy scenario that leads to a positive sign for $\beta$ [32,33].

A positive GUP parameter $\beta$ is apparent in the thought experiment of observing an electron through a Heisenberg microscope [34]. The additional uncertainty in position of the electron due to gravitational interaction with the photon turns out to be a positive quantity, of the order of $\ell_P^2\Delta p/\hbar$ [35], implying $\beta$ is positive. Similarly, a string theoretic consideration with a length scale $\ell_*$ also leads to the same additional uncertainty in position with $\ell_*$ replacing $\ell_P$ [32,33]. In addition, measurement of the radius of an extremal black hole by dropping a photon into it and by observing the re-emitted photon gives a similar (positive) estimate for the uncertainty [36–38].

Moreover, it has been shown that a negative GUP parameter $\beta$ gives rise to an unphysical mass–radius relation for white dwarfs [5]. Consequently, we include the effect of quantum gravity on white dwarfs via the GUP with a positive sign for $\beta$. However, this poses the well-known problem that the Chandrasekhar limit ceases to exist. It was in fact suggested in [5] that a consistent solution of the problem could be obtained within the framework of general relativity. Since white dwarfs respect the Chandrasekhar limit, it is extremely important to solve this problem posed by GUP. A satisfactory model of white dwarfs ought to be based on a rigorous treatment of the gravitational field so that the gravitational collapse for a sufficiently massive white dwarf is well represented.

In this paper, we present a complete and rigorous approach to resolve this problem. We take the framework of general relativity (GR) and calculate the stellar structure of white dwarfs for positive GUP parameter $\beta$. We carry out a dynamical stability analysis of the equilibrium configurations so that the maximal stable configuration is identified. In this framework, we rigorously confirm the existence of Chandrasekhar's limit within the electroweak upper bound [39] of the GUP parameter $\beta$. More precisely, we find that the Chandrasekhar limit robustly exists even when the value of $\beta$ is made four orders higher than the electroweak bound.

The remainder of the paper is organized as follows. In §2, we present the Fermionic equation of state following from GUP. In §3, we give details of the mass–radius relation in the framework of general relativity. Section 4 presents the dynamical stability analysis confirming the Chandrasekhar limit. A discussion and conclusion is presented in §5.

## 2. Generalized uncertainty principle and Fermionic equation of state

A minimum uncertainty in length due to the granular structure of space, which is essentially a quantum gravitational effect, can be incorporated by generalizing the Heisenberg commutation relations [25] to

$$
\left.
\begin{aligned}
[\hat{x}_i,\ \hat{p}_j] &= i\hbar\,\delta_{ij}(1 + \beta\hat{\mathbf{p}}^2), \\
[\hat{p}_i,\ \hat{p}_j] &= 0 \\
[\hat{x}_i,\ \hat{x}_j] &= 2i\hbar\,\beta(\hat{p}_i\hat{x}_j - \hat{p}_j\hat{x}_i).
\end{aligned}
\right\}
\tag{2.1}
$$

and

These generalized commutation relations incorporate a modified high momentum behaviour via the terms containing $\beta \sim \ell_P^2/\hbar^2$, where $\ell_P = \sqrt{G\hbar/c^3} = 1.6162 \times 10^{-33}$ cm is the Planck length. Considering a classical Liouville's equation, it was shown [40] that the invariant measure of the phase volume takes up a factor of $(1 + \beta\mathbf{p}^2)^{-3}$. This imposes a severe restriction on the allowed quantum states and thus modifies the thermodynamic properties with respect to the ideal case.

The inclusion of quantum gravitational fluctuations via the generalized uncertainty principle in the equation of state of a degenerate electron gas was studied earlier in different contexts [19,41–44]. In this section, we present the number density $n$, pressure $P$ and energy density $\varepsilon$ of the electron degenerate gas. With the modified phase volume, we employ the standard method of statistical mechanics to the relativistic electron gas assuming $T = 0$, yielding

$$
n = \frac{8\pi}{h^3}\int_0^{p_F}\frac{p^2\,\mathrm{d}p}{(1 + \beta p^2)^3}
\tag{2.2}
$$

and

$$
P = \frac{8\pi}{h^3}\int_0^{p_F}\frac{p^2\,\mathrm{d}p}{(1 + \beta p^2)^3}(E_F - E_{\mathbf{p}}),
\tag{2.3}
$$

leading to

$$
n(\xi) = \frac{K}{m_e c^2}\tilde{n}(\xi)\quad\text{and}\quad P(\xi) = K\tilde{P}(\xi),
\tag{2.4}
$$

where

$$
\tilde{n}(\xi) = \frac{1}{\alpha^3}\left[\tan^{-1}(\alpha\xi) - \frac{\alpha\xi(1 - \alpha^2\xi^2)}{(1 + \alpha^2\xi^2)^2}\right]
\tag{2.5}
$$

and

$$
\tilde{P}(\xi) = \frac{\sqrt{1 + \xi^2}}{\alpha^3}\left\{\tan^{-1}(\alpha\xi) - \frac{\alpha\xi}{(1 - \alpha^2)(1 + \alpha^2\xi^2)}\right\} + \frac{1}{(1 - \alpha^2)^{3/2}}\tanh^{-1}\frac{\xi\sqrt{1 - \alpha^2}}{\sqrt{1 + \xi^2}}
\tag{2.6}
$$

with $\xi = p_F/m_e c$, $p_F$ being the Fermi momentum, $\alpha = \beta m_e^2 c^2 = \beta_0 m_e^2/M_P^2$ ($M_P = \sqrt{\hbar c/G} = 2.1765 \times 10^{-5}$ g) and $K = \pi m_e^4 c^5/h^3$.

The internal kinetic energy $\varepsilon_{\text{int}}(\xi)$ of the electron gas (for $T = 0$) is given by

$$
\varepsilon_{\text{int}}(\xi) = \frac{8\pi}{h^3}\int_0^{p_F}\frac{p^2\,\mathrm{d}p}{(1 + \beta p^2)^3}\left\{\sqrt{p^2 c^2 + m_e^2 c^4} - m_e c^2\right\}.
\tag{2.7}
$$

In the dimensionless quantities, the above equation becomes

$$
\varepsilon_{\text{int}}(\xi) = \frac{8\pi m_e^4 c^5}{h^3}\int_0^{\xi}\frac{\xi'^2\,\mathrm{d}\xi'}{(1 + \alpha^2\xi'^2)^3}\left\{\sqrt{\xi'^2 + 1} - 1\right\},
\tag{2.8}
$$

leading to

$$
\epsilon_{\text{int}}(\xi) = \left\{\frac{\xi\sqrt{1 + \xi^2}[1 + (2 - \alpha^2)\xi^2]}{(1 - \alpha^2)(1 + \alpha^2\xi^2)^2} - \frac{1}{(1 - \alpha^2)^{3/2}}\tanh^{-1}\frac{\xi\sqrt{1 - \alpha^2}}{\sqrt{1 + \xi^2}}\right\} - \tilde{n}.
\tag{2.9}
$$

The rest mass density $\rho_0(\xi) = m_u \mu_e\, n(\xi)$ is related to the energy density as $\varepsilon(\xi) = \rho_0(\xi)c^2 + \varepsilon_{\text{int}}(\xi)$, where $m_u = 1.6605 \times 10^{-24}$ g is the atomic mass unit and $\mu_e = A/Z$, with $A$ the mass number and $Z$ the atomic number. Thus, the energy density

$$\varepsilon(\xi) = \frac{K}{q}\tilde{\varepsilon}(\xi), \tag{2.10}$$

where $q = m_e/\mu_e\, m_u$ and the dimensionless energy density $\tilde{\varepsilon}(\xi)$ is given by

$$\tilde{\varepsilon}(\xi) = (1-q)\tilde{n} + q\left\{ \frac{\xi\sqrt{1+\xi^2}[1 + (2-\alpha^2)\xi^2]}{(1-\alpha^2)(1+\alpha^2\xi^2)^2} - \frac{1}{(1-\alpha^2)^{3/2}}\tanh^{-1}\frac{\xi\sqrt{1-\alpha^2}}{\sqrt{1+\xi^2}} \right\}, \tag{2.11}$$

in the high Fermi momentum limit, that is $\xi \to \infty$,

$$\tilde{n}(\xi) \longrightarrow \frac{\pi}{2\alpha^3} = k_1, \tag{2.12}$$

$$\tilde{P}(\xi) \longrightarrow k_1\xi - k_2 \tag{2.13}$$

and
$$\tilde{\varepsilon} \longrightarrow k_1(1-q) + qk_2 = 3\kappa \tag{2.14}$$

with

$$k_2 = \frac{1}{\alpha^4}\frac{(2-\alpha^2)}{(1-\alpha^2)} - \frac{\tanh^{-1}\sqrt{1-\alpha^2}}{(1-\alpha^2)^{3/2}}, \tag{2.15}$$

where $k_1$, $k_2$ and $\kappa$ are constants. These high momentum limits are drastically different from the ideal case due to the role of the generalized uncertainty principle.

Moreover, the relativistic adiabatic index $\gamma$ for the degenerate electron gas is obtained as

$$\gamma = \frac{\varepsilon + P}{P}\left(\frac{\mathrm{d}P}{\mathrm{d}\varepsilon}\right)_s = \frac{1}{8}\left(\frac{\tilde{n}^2}{\tilde{P}}\right)\frac{(1+\alpha^2\xi^2)^3}{\xi\sqrt{1+\xi^2}}, \tag{2.16}$$

so that $\gamma \to (\pi/16)\,\alpha^3$ in the limit $\xi \to \infty$, unlike the ideal case ($\gamma_{\text{ideal}} = 4/3$).

# 3. Mass–radius relation

We study mass–radius relation of the equilibrium configurations in the framework of general relativity in this section. For the matter interior to the star, the equilibrium values of the pressure $P(r)$ and energy density $\varepsilon(r)$ are therefore determined by the Tolman–Oppenheimer–Volkoff (TOV) equations [45,46]

$$\frac{\mathrm{d}P}{\mathrm{d}r} = -\frac{G}{c^2 r}(\varepsilon + P)\frac{(m + 4\pi P r^3/c^2)}{(r - 2Gm/c^2)} \tag{3.1}$$

with

$$\frac{\mathrm{d}m}{\mathrm{d}r} = \frac{4\pi}{c^2}\varepsilon r^2. \tag{3.2}$$

It may be observed that the equation of state is in a parametric form where the Fermi momentum $p_F$ of the electron degenerate gas occurs in the expressions for pressure and energy density given by equations (2.4), (2.6), (2.10) and (2.11). We express the TOV equations (3.1) and (3.2) in terms of the dimensionless quantities $\xi = p_F/m_e c$, $v = m/m_0$ and $\eta = r/r_0$, where $m_0 = (qc^2)^2/G^{3/2}\sqrt{4\pi K}$ and $r_0 = (qc^2)/\sqrt{4\pi GK}$. Thus we obtain

$$\frac{\mathrm{d}\xi}{\mathrm{d}\eta} = -\frac{1}{\eta}\frac{\sqrt{1+\xi^2}}{\xi}\left(1 - q + q\sqrt{1+\xi^2}\right)\frac{v + q\tilde{P}\eta^3}{\eta - 2qv} \tag{3.3}$$

and

$$\frac{\mathrm{d}v}{\mathrm{d}\eta} = \tilde{\varepsilon}\eta^2. \tag{3.4}$$

## 3.1. Asymptotic solutions

For a preliminary idea about the mass–radius relation, we study the asymptotic solutions of the TOV equations in the low- and high-Fermi momentum limits.

### 3.1.1. Low momentum limit, $\xi \to 0$

For low values of $\xi$, it can be shown that equations (3.3) and (3.4) reduce to

$$\xi \frac{d\xi}{d\eta} = -\frac{v}{\eta^2} \tag{3.5}$$

and

$$\frac{dv}{d\eta} = \frac{8}{3} \xi^3 \eta^2. \tag{3.6}$$

which can be combined to form a second-order differential equation, given by

$$\frac{3}{16} \frac{1}{\eta^2} \frac{d}{d\eta} \left( \eta^2 \frac{d\xi^2}{d\eta} \right) + \xi^3 = 0. \tag{3.7}$$

Defining $\xi^2(\eta)/\xi_c^2$ as $\theta(\zeta)$, with $\xi_c$ the central dimensionless Fermi momentum, and $\zeta$ a new dimensionless coordinate, $\zeta = \sqrt{16\xi_c/3}\eta$, we reduce the above equation to

$$\frac{1}{\zeta^2} \frac{d}{d\zeta} \left( \zeta^2 \frac{d\theta}{d\zeta} \right) + \theta^{3/2} = 0, \tag{3.8}$$

which is the Lane–Emden equation of index 3/2. The numerical solution for this differential equation is given in Weinberg [47]. For the boundary conditions $\theta(0) = 1$ and $\theta'(0) = 0$, one can immediately obtain the radius of the white dwarf as

$$R = \sqrt{\frac{3}{16\xi_c}} r_0 \zeta_R, \tag{3.9}$$

where $\zeta_R = 3.65375$ is the first zero of the Lane–Emden function $\theta(\zeta)$ of index 3/2.

Similarly, the asymptotic behaviour of the mass of the white dwarf can be obtained from the integral expression of equation (3.2), namely,

$$M = \frac{4\pi}{c^2} \int_0^R \varepsilon(r) r^2 \, dr = \frac{4\pi}{c^2} \frac{A}{q} \frac{8}{3} \int_0^R \xi^3 r^2 \, dr. \tag{3.10}$$

We rewrite this equation in the new dimensionless variable $\zeta$, yielding

$$M = \sqrt{\frac{3\xi_c^3}{64}} m_0 \int_0^{\zeta_R} \theta^{3/2} \zeta^2 \, d\zeta, \tag{3.11}$$

thus obtaining the mass of the white dwarf as

$$M = -\sqrt{\frac{3\xi_c^3}{64}} m_0 \zeta_R^2 \left( \frac{d\theta}{d\zeta} \right)_{\zeta = \zeta_R}. \tag{3.12}$$

The value of $(-\zeta^2 \, d\theta/d\zeta)_{\zeta = \zeta_R}$ is 2.71406 [47].

Thus, the above asymptotic analysis predicts that $R \sim \xi_c^{-1/2}$ and $M \sim \xi_c^{3/2}$, giving the mass–radius relation $R \sim M^{-1/3}$, implying that the radius decreases as the mass increases.

It is important to note that these expressions of mass and radius are independent of the GUP parameter $\alpha$ (or, equivalently, $\beta$). Thus for low mass white dwarfs the GUP has an insignificant effect on the mass–radius relation and we expect that the mass–radius curve would coincide with that of Chandrasekhar's for low values of central Fermi momentum $\xi_c$ (or, equivalently, low central density $\rho_c$).

### 3.1.2. High momentum limit, $\xi \to \infty$

For high values of $\xi$, the TOV equations reduce to

$$\frac{d\xi}{d\eta} = -\frac{k_1}{3} \frac{\eta}{1 - 2q\kappa\eta^2} (1 - q + q\xi) \left( 1 - q + 3q\xi - 2q\frac{k_2}{k_1} \right) \tag{3.13}$$

and

$$v = \kappa\eta^3. \tag{3.14}$$

Since typically $\alpha \sim 0.1$, the ratio $k_2/k_1 \sim (4/\pi\alpha)$; hence the last term in the second brackets can be ignored if $\alpha\xi \gg 8/3\pi$. Since we are looking for the solutions with $\xi \to \infty$, we shall ignore this term, obtaining

$$\frac{d\xi}{d\eta} = -\frac{k_1}{3}\frac{\eta}{1-2q\kappa\eta^2}(1+q\xi)(1+3q\xi),$$ (3.15)

where we have used the fact that $q \sim 10^{-4}$. The solution of the above equation is given by

$$\frac{1+3q\xi}{1+q\xi} = (1-2q\kappa\eta^2)^{-k_1/6\kappa} + \text{const.}$$ (3.16)

Using the boundary conditions, we can immediately obtain the integration constant and hence the radius of the star as

$$\eta_R = \frac{1}{\sqrt{2q\kappa}}\left\{1 - \left(\frac{1+q\xi_c}{1+3q\xi_c}\right)^{6\kappa/k_1}\right\}^{1/2}.$$ (3.17)

Since $6\kappa/k_1 \approx 2$, we have

$$\eta_R = \frac{1}{\sqrt{2q\kappa}}\left\{1 - \left(\frac{1+q\xi_c}{1+3q\xi_c}\right)^{2}\right\}^{1/2}.$$ (3.18)

Thus from equation (3.14) the mass becomes

$$v_R = \left(\frac{1}{2q}\right)^{3/2}\frac{1}{\sqrt{\kappa}}\left\{1 - \left(\frac{1+q\xi_c}{1+3q\xi_c}\right)^{2}\right\}^{3/2}.$$ (3.19)

As the central Fermi momentum approaches larger and larger values, we see that the radius and mass approach maximum values, given by

$$R_{\max} = \frac{2}{3}\frac{r_0}{\sqrt{q\kappa}} \quad \text{and} \quad M_{\max} = \frac{8}{27}\frac{m_0}{\sqrt{\kappa}q^{3/2}}.$$ (3.20)

## 3.2. Exact solutions

In this section, we obtain exact solutions of the TOV equations (3.3) and (3.4) employing the GUP equation of state expressed by equations (2.6) and (2.11) in parametric form. The numerical integrations are carried out with the boundary conditions $\xi(0) = \xi_c$, $v(0) = 0$ and $\xi(\eta_R) = 0$, where $\eta_R$ denotes the dimensionless radius of the star. The resulting mass–radius relations for different strengths of the dimensionless GUP parameter $\beta_0$ are shown in figures 1 and 2.

It is apparent from figures 1 and 2 that, for large values of $\beta_0$, the mass–radius relations given by the GUP equation of state deviate significantly from the ideal case, whereas for smaller values of $\beta_0$, such deviations are smaller.

In figure 1a,b, we display the mass–radius curves for higher magnitudes of the GUP parameter such as $\beta_0 = 10^{44}$, $10^{42}$, $10^{41}$ and $10^{40}$. We see that the mass–radius curves coincide with Chandrasekhar's curve only for low values of the central Fermi momentum $\xi_c$, as shown in the right-hand part of the inset in figure 1b. This is evident from the fact that the TOV equation reduces to Newtonian equation in the low density regime. Moreover, we see from the right-hand part of figure 1a that all curves nearly coincide irrespective of the strength of the GUP parameter $\beta_0$. This is due to the fact that $\beta_0$ disappears from the asymptotic equations in this regime as we have seen above in §3.1.1 in the low momentum limit.

For higher $\xi_c$ values, the exact mass–radius curve reaches a point where the radius is minimum $R_{\min}$. The $R_{\min}$ value is smaller for smaller $\beta_0$ values as seen in figure 1a. On further increasing $\xi_c$, both the mass and radius increase reaching terminal values as shown in figure 1b denoted by an open circle. In this regime, we see that the analytically obtained high momentum solution (in §3.1.2) coincides with the exact mass–radius curve as shown in figure 1b. Moreover, the terminal values of radius $R_{\max}$ and mass $M_{\max}$ given by equation (3.20) are found to be nearly the same as given by the exact solutions. However, these terminal values are excessively high, as evident from figure 1b.

Exact mass–radius curves for intermediate strengths of the GUP parameter $\beta_0$ (in the range $4.50 \times 10^{39} \le \beta_0 \le 6.30 \times 10^{39}$) are shown in figure 2a. We see a cross-over in the behaviour of the curves

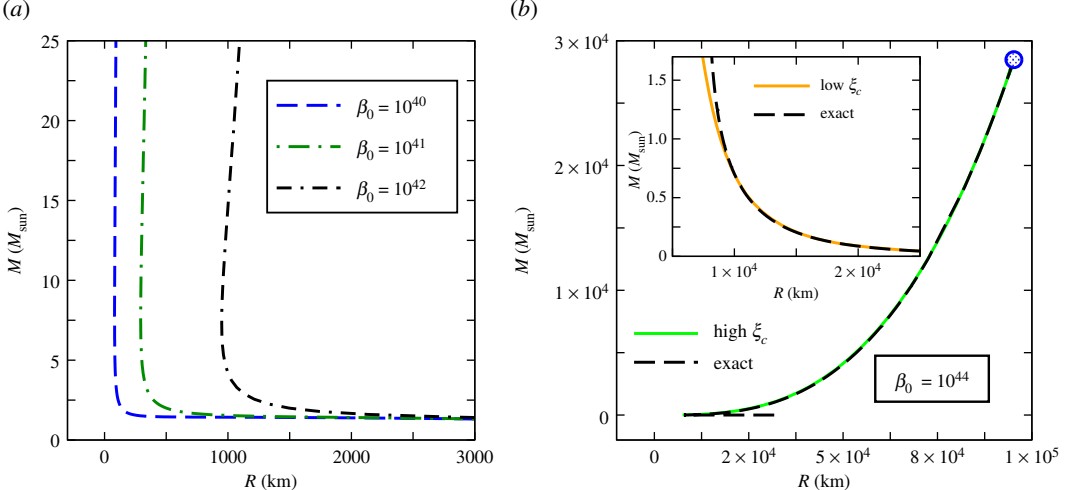

**Figure 1.** (a) Exact mass–radius relations for white dwarfs with GUP equation of state for $\beta_0 = 10^{42}$, $10^{41}$ and $10^{40}$. (b) Exact mass–radius relations (dashed curves) for $\beta_0 = 10^{44}$ in comparison with the corresponding analytically obtained asymptotic solution (smooth curve) given by equations (3.9) and (3.12) in the high $\xi_c$ limit. The open circle represents the maximum values of mass $M_{max}$ and radius $R_{max}$. The lower left region of the exact mass–radius curve is blown up (dashed curve) in the inset where it is compared with the analytically obtained asymptotic solution in the low $\xi_c$ limit (smooth curve).

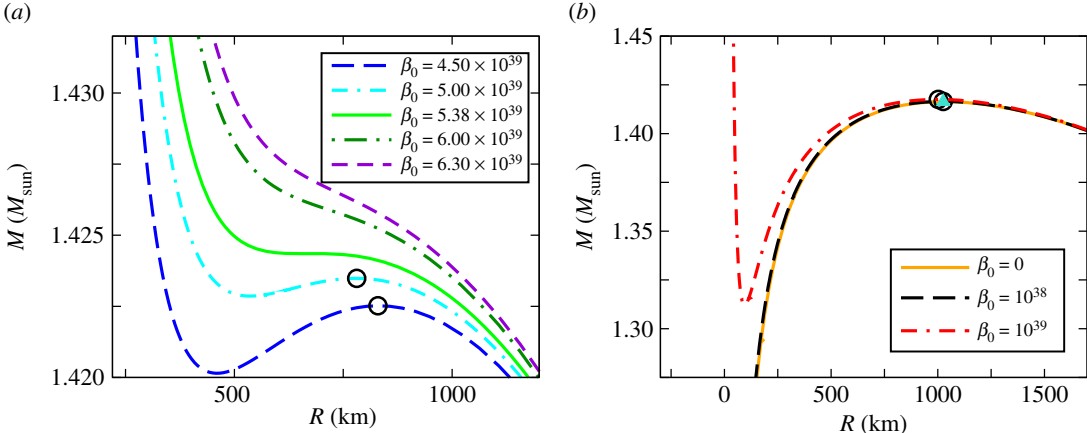

**Figure 2.** (a) Exact mass–radius relations for $\beta_0$ in the range $4.50 \times 10^{39} \leq \beta_0 \leq 6.30 \times 10^{39}$. The mass–radius relation for $\beta_0 = \bar{\beta}_0 = 5.38 \times 10^{39}$ demarcates these curves into two classes. For $\beta_0 > \bar{\beta}_0$, there is no maximal point, whereas for $\beta_0 < \bar{\beta}_0$ maximal points $(R^*, M^*)$ exist (shown by open circles). (b) Exact mass–radius relations for $\beta_0 = 10^{39}$ and $10^{38}$ in comparison with that of the ideal case, $\beta_0 = 0$. Proximity of the maximal points $(R^*, M^*)$ are shown by open circles (for $\beta_0 = 10^{39}$ and $10^{38}$) with that of the ideal case, shown as a solid triangle ($\beta_0 = 0$).

around the value $\beta_0 = \bar{\beta}_0 = 5.38 \times 10^{39}$. For $\beta_0 > \bar{\beta}_0$, the mass–radius curves do not have a maximal point, whereas for $\beta_0 < \bar{\beta}_0$, there exist maximal points. Figure 2b compares the mass–radius relation for smaller values of $\beta_0$ (=$10^{39}$ and $10^{38}$) with the ideal case ($\beta_0 = 0$). We see that the maxima of the mass–radius curves for these values of $\beta_0$ nearly coincide with the maxima of the ideal case. It is also important to note that the maxima shifts slightly towards the right (figure 2a) as the value of $\beta_0$ is decreased until the maxima coincide with the ideal value (figure 2b).

A more rigorous treatment is required to assert whether these maxima correspond to the onset of gravitational instability. Although Newtonian gravity gives the stellar structure of low-mass white dwarfs in the ideal case (with $\beta = 0$), the correct mass–radius curve and the dynamical instability for high-mass white dwarfs is determined by general relativity. Consequently, it is critical to analyse the role of the GUP parameter in determining the dynamical instability of white dwarfs. In the following section, we perform a rigorous stability analysis of the equilibrium configurations by investigating the dynamical instability in the framework of general relativity. It consists of studying the dynamics of time dependent infinitesimal radial perturbations about the equilibrium configuration at

every point inside the star in a homologous manner [48]. The time evolution of these perturbations determined by the central Fermi momentum $\xi_c$ and the GUP parameter $\beta_0$ establishes whether the system is stable or otherwise.

# 4. Dynamical stability analysis

As we have already noted, dynamical stability analysis consists of the investigation of the time evolution of homologous infinitesimal perturbations about the equilibrium configuration [48–51]. The corresponding metric interior to the star is expressed as

$$ds^2 = e^{\nu + \delta\nu} c^2 \, dt^2 - e^{\mu + \delta\mu} \, dr^2 - r^2 (d\theta^2 + \sin^2\theta \, d\phi^2), \tag{4.1}$$

where $\nu(r)$ and $\mu(r)$ are the equilibrium metric potentials and the perturbations $\delta\nu(r, t)$ and $\delta\mu(r, t)$ are due to small radial Lagrangian displacements $\zeta(r, t)$. This induces perturbations $\delta P(r, t)$ and $\delta\varepsilon(r, t)$ to the equilibrium pressure $P(r)$ and energy density $\varepsilon(r)$. The smallness of the perturbation allows one to consider sinusoidal displacements $\zeta(r, t) = r^{-2} e^{\nu/2} \psi(r) e^{i\omega t}$. The corresponding equation for the radial oscillation can be obtained in the Strum–Liouville form [52]

$$\frac{d}{dr}\left( U \frac{d\psi}{dr} \right) + \left( V + \frac{\omega^2}{c^2} W \right) \psi = 0, \tag{4.2}$$

satisfying the boundary conditions $\psi = 0$ at $r = 0$ and the Lagrangian change in pressure $\delta P = -e^{\nu/2}(\gamma P / r^2)$ $d\psi/dr = 0$ at $r = R$. In the above equation,

$$U(r) = e^{(\mu + 3\nu)/2} \frac{\gamma P}{r^2}, \tag{4.3}$$

$$V(r) = -4 \frac{e^{(\mu+3\nu)/2}}{r^3} \frac{dP}{dr} - \frac{8\pi G}{c^4} \frac{e^{3(\mu+\nu)/2}}{r^2} P(P + \varepsilon) + \frac{e^{(\mu+3\nu)/2}}{r^2} \frac{1}{P + \varepsilon} \left( \frac{dP}{dr} \right)^2 \tag{4.4}$$

and

$$W(r) = \frac{e^{(3\mu+\nu)/2}}{r^2} (P + \varepsilon), \tag{4.5}$$

with the adiabatic index $\gamma$, given by

$$\gamma = \frac{\varepsilon + P}{P} \left( \frac{dP}{d\varepsilon} \right)_s. \tag{4.6}$$

Integrating equation (4.2) upon left-multiplying by $\psi$, one obtains the integral

$$J[\psi] = \int_0^R \left\{ U\psi'^2 - V\psi^2 - \frac{\omega^2}{c^2} W\psi^2 \right\} dr, \tag{4.7}$$

where $\psi' = d\psi/dr$, and the boundary conditions eliminate the surface term. It can be shown that equation (4.2) is reproduced from the variational principle $\delta J[\psi] = 0$. Thus, the lowest characteristic eigenfrequency of the normal mode is obtained from

$$\frac{\omega_0^2}{c^2} = \min_{\psi(r)} \frac{\int_0^R \{U\psi'^2 - V\psi^2\} \, dr}{\int_0^R W\psi^2 \, dr}. \tag{4.8}$$

The star remains in stable equilibrium so long as this equation yields positive values for $\omega_0^2$. On the other hand, a negative $\omega_0^2$ signifies unstable equilibrium. A power series solution of equation (4.2) about $r = 0$ gives $\psi(r) \propto r^3$ in the leading order for which $\zeta(r)$ and $\zeta'(r)$ are finite. A good approximation for the trial function of the fundamental mode can be taken as the simple form $\psi(r) = c_0 r^3$ [48,49,53]. With this choice, the onset of instability, hence the critical density $\rho_c^*$ for gravitational collapse, can be identified with a zero eigenfrequency solution of equation (4.8).

For the matter interior to the star, the equilibrium values of the pressure $P(r)$ and energy density $\varepsilon(r)$ are determined by the Tolman–Oppenheimer–Volkoff (TOV) equations (3.1) and (3.2) and the interior Schwarzschild metric potentials satisfying Einstein's field equations are given by [45,46]

$$e^{-\mu(r)} = 1 - \frac{2Gm}{c^2 r} \tag{4.9}$$

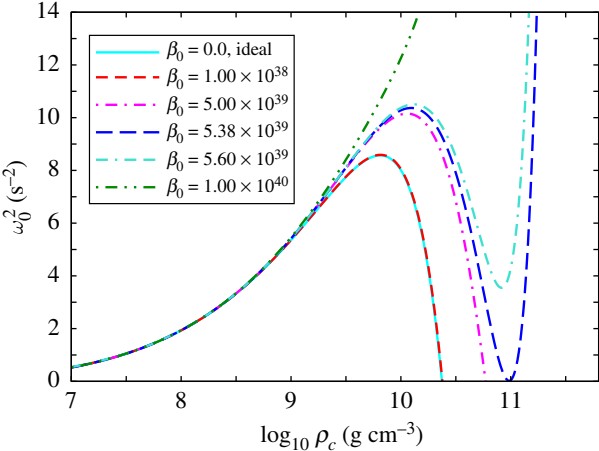

**Figure 3.** Eigenfrequency of the fundamental mode $\omega_0^2$ against central density $\rho_c$ for various values of the GUP parameter $\beta_0$.

and

$$e^{\nu(r)} = \left(1 - \frac{2GM}{c^2 R}\right) \exp\left[-2 \int_0^{P(r)} \frac{dP}{\varepsilon + P}\right]. \tag{4.10}$$

## 4.1. Eigenfrequency of the fundamental mode

The interior Schwarzschild metric potentials can be written in the dimensionless variables as

$$e^{-\mu(\eta)} = 1 - 2q\frac{\upsilon}{\eta} \tag{4.11}$$

and

$$e^{\nu(\eta)} = \left(1 - 2q\frac{\upsilon_R}{\eta_R}\right)\left(\frac{1}{1 - q + q\sqrt{1 + \xi^2}}\right)^2, \tag{4.12}$$

where the expression for $e^\nu$ is obtained from the equation of state given by equations (2.6) and (2.11).

The solution of the TOV equations (3.3) and (3.4), and equations (4.11) and (4.12) give all quantities necessary for the evaluation of the functions $U(r)$, $V(r)$ and $W(r)$ in equations (4.3)–(4.5). We may rewrite equation (4.8) in dimensionless form as

$$\omega_0^2 = \left(\frac{qc^2}{r_0^2}\right)\frac{\mathcal{I} + \mathcal{J}}{\mathcal{K}}, \tag{4.13}$$

where

$$\mathcal{I} = \int_0^{\eta_R} e^{(\mu + 3\nu)/2} \frac{\gamma\tilde{P}}{\eta^2} \psi'^2 \, d\eta, \tag{4.14}$$

$$\mathcal{J} = \int_0^{\eta_R} \frac{e^{(\mu + 3\nu)/2}}{\eta^2} \left[\frac{4}{\eta}\frac{d\tilde{P}}{d\eta} + 2q\, e^{\mu}\tilde{P}(\tilde{\varepsilon} + q\tilde{P}) - \frac{q}{\tilde{\varepsilon} + q\tilde{P}}\left(\frac{d\tilde{P}}{d\eta}\right)^2\right]\psi^2 \, d\eta \tag{4.15}$$

and

$$\mathcal{K} = \int_0^{\eta_R} e^{(3\mu + \nu)/2} \frac{\tilde{\varepsilon} + q\tilde{P}}{\eta^2}\psi^2 \, d\eta. \tag{4.16}$$

We thus numerically evaluate the integrals in equations (4.14)–(4.16) with the trial function $\psi = c_0\eta^3$, with $c_0$ a disposable constant, for different choices of the GUP parameter $\beta_0$. Consequently, we obtain the eigenfrequency of the fundamental mode $\omega_0^2$ from equation (4.13). As stated earlier, stable configurations correspond to positive values of $\omega_0^2$, whereas a zero frequency solution indicates the onset of a dynamical instability signifying the onset of a gravitational collapse.

We display the results of the numerical integrations in figure 3, where the eigenfrequency $\omega_0^2$ is plotted with respect to the central density $\rho_c$ (= $\varepsilon_c/c^2$) for different values of $\beta_0$. For low mass white dwarfs with central densities $\rho_c \lesssim 10^9\,\mathrm{g\,cm}^{-3}$, we observe that the pulsation frequencies overlap signifying the irrelevance of the effect of GUP in this range of central densities. The

**Table 1.** Critical values of the central density $\rho_c^*$, mass $M^*$, and radius $R^*$ for different values of the GUP parameter $\beta_0$ at the onset of dynamical instability determined by the vanishing eigenfrequency of the fundamental mode.

| $\beta_0$ | $\rho_c^*$ (g cm$^{-3}$) | $M^*$ (M$_\odot$) | $R^*$ (km) |
|---|---|---|---|
| $5.38 \times 10^{39}$ | $1.0105 \times 10^{11}$ | 1.4244 | 655.5629 |
| $5.00 \times 10^{39}$ | $5.8618 \times 10^{10}$ | 1.4235 | 776.3669 |
| $1.00 \times 10^{38}$ | $2.3801 \times 10^{10}$ | 1.4165 | 1021.6162 |
| $1.00 \times 10^{34}$ | $2.3588 \times 10^{10}$ | 1.4164 | 1024.3821 |

pulsation frequencies start to deviate from each other in the higher density regime depending on the value of $\beta_0$.

For $\beta_0 \leq \bar{\beta}_0 = 5.38 \times 10^{39}$, there exist zero eigenfrequency solutions at central densities $\rho_c^*$, suggesting the onset of gravitational collapse. The existence of imaginary eigenfrequency solution corresponding to unstable configuration is possible only for $\beta_0 < \bar{\beta}_0$. For $\beta_0 > \bar{\beta}_0$, zero eigenfrequency solutions are not possible even for arbitrarily high central densities $\rho_c$, signifying stability of excessively massive white dwarfs. We also see that the curve for $\beta_0 = 10^{38}$ nearly coincides with that for the ideal case ($\beta_0 = 0$). This means that all curves in the range $0 \leq \beta_0 \leq 10^{38}$ overlap (to a good approximation) giving rise to approximately the same onset density $\rho_c^*$ for gravitational collapse. A legitimate upper bound is given by the electroweak limit $\beta_0 \sim 10^{34}$ [39] which is well within the range $0 \leq \beta_0 \leq 10^{38}$. Since this onset density is nearly $2.3588 \times 10^{10}$ g cm$^{-3}$, Chandrasekhar's general relativistic mass $\sim 1.42$ M$_\odot$ is easily recovered in this range which extends four orders of magnitude beyond the electroweak bound.

The above discussions lead to parallel observations from figure 2a where $\beta_0 = \bar{\beta}_0$ demarcates a change in behaviour of the mass–radius curves. The non-existence of a maximal point in the mass–radius curve for $\beta_0 > \bar{\beta}_0$ is evident from the fact that there exists no critical density $\rho_c^*$ corresponding to a zero eigenfrequency solution. On the other hand, for $\beta_0 < \bar{\beta}_0$, the existence of maximal points $(R^*, M^*)$ in the mass–radius curves are consequences of the zero eigenfrequency solutions at $\rho_c^*$. The branches towards the right of the maximal point $(R^*, M^*)$ correspond to lower central densities $\rho_c < \rho_c^*$ and thus the stability of this branch is confirmed by the fact that $\omega_0^2$ is positive, as shown in figure 3. On the other hand, the branches towards the left of the maximal point $(R^*, M^*)$ correspond to instability as $\omega_0^2$ becomes negative (not shown in figure 3) and they correspond to $\rho_c > \rho_c^*$.

As $\beta_0$ is deceased towards $10^{38}$, the maximal points $(R^*, M^*)$ approach closer to each other and nearly coincide at $\beta_0 = 10^{38}$. The corresponding critical values are displayed in table 1 where it is evident that the critical mass approaches the limit 1.416 M$_\odot$ and the radius 1024 km.

Thus in addition to asserting the existence of the Chandrasekhar limit, the stability analysis confirms the fact that the radius decreases as the mass increases for stable configurations of white dwarfs.

# 5. Discussion and conclusion

There have been a few recent attempts to restore the Chandrasekhar limit when white dwarfs are described by GUP-enhanced equation of state. As we have discussed earlier, there are various scenarios [26–28] pointing to the possibility of $\beta$ being negative. As shown in [5], a choice of a negative GUP parameter $\beta$ gives rise to the mass–radius relation

$$R = \sqrt{|\beta|} \frac{M m_e^{1/3} c}{\sqrt{M_{Ch}^{2/3} - M^{2/3}}} \ell_P, \tag{5.1}$$

in the relativistic limit, giving the Chandrasekhar mass $M_{Ch}$ as an upper bound. However, this mass–radius relation has inconsistencies with observations, namely, (i) as the mass $M$ increases, the radius $R$ also increases, and (ii) the radius diverges as the mass approaches the Chandrasekhar limit, preventing the formation of more compact objects as the density would be infinitely diluted. In fact, observations indicate that the radius decreases with the increase in mass of white dwarfs. Moreover, we expect the formation of highly dense objects such as neutron stars or black holes when the mass exceeds the Chandrasekhar limit. These inconsistencies do not appear when we take $\beta$ to be a positive quantity.

In an alternative approach to circumvent the problem of non-existence of the Chandrasekhar mass, an extended GUP [6] was suggested by incorporating the effect of cosmological constant $\Lambda$, so that

$$\Delta x \Delta p \geq \frac{\hbar}{2} \left\{ 1 + \beta(\Delta p)^2 - \lambda \frac{(\Delta x)^2}{L_\Lambda^2} \right\}, \tag{5.2}$$

with $L_\Lambda^2 = \lambda / \Lambda$ which is positive for de-Sitter expansion of the Universe ($\lambda = +3$). Although the observed value of $\Lambda$ is very small, namely $\Lambda \sim 10^{-52}$ m$^{-2}$, they showed that this reformulation of GUP leads to a mass–radius relation whose physically acceptable solution is strongly dominated by the cosmological terms and the contribution from $\beta$ is insignificant, making the sign of $\beta$ irrelevant. This mass–radius relation clearly shows that the Chandrasekhar mass is the upper bound. However, this mass–radius relation also suffers from the same inconsistencies as described above.

Because of these inconsistencies, it is essential to resolve the issue of non-existence of Chandrasekhar's limit in a cogent fashion so that all assumptions in the theory lead to results in agreement with observations. We therefore formulated the problem in a rigorous manner by adopting general relativity vis-à-vis GUP-enhanced equation of state with the assumption of a positive GUP parameter $\beta$. Importantly, we find that the Chandrasekhar mass is assured for $\beta_0$ values below $\bar{\beta}_0$, due to the onset of gravitational collapse. We also note that the electroweak upper bound for $\beta_0$ is much below $\bar{\beta}_0$ so that physical existence of Chandrasekhar's limit is guaranteed.

The above conclusion stems from a rigorous stability analysis of the equilibrium configurations as displayed in figure 3, where the eigenfrequency of the fundamental mode $\omega_0^2$ is plotted with respect to the central density $\rho_c$ for different values of $\beta_0$. We see that a vanishing eigenfrequency exists when $\beta_0 \leq 5.38 \times 10^{39} = \bar{\beta}_0$, giving rise to a dynamical instability at critical central densities $\rho_c^*$. However, for $\beta_0 > \bar{\beta}_0$, no dynamical instability occurs because of the nonexistence of a zero eigenfrequency solution, implying that these configurations remain stable for arbitrarily high values of $\rho_c$ leading to excessively massive white dwarfs. However, these solutions are physically unacceptable because the corresponding $\beta_0$ values are well above the electroweak bound.

An important point to observe from figure 3 is that the eigenfrequencies for $\beta_0 = 10^{38}$ practically coincide with those of the ideal case, $\beta_0 = 0$. Thus in the range $0 < \beta_0 < 10^{38}$, the critical density $\rho_c^*$ for the onset of gravitational collapse (determined by the vanishing eigenfrequency) remains practically unaltered. We find $\rho_c^* = 2.3801 \times 10^{10}$ g cm$^{-3}$ for $\beta_0 = 10^{38}$, which is nearly the same as Chandrasekhar's critical value of $2.3 \times 10^{10}$ g cm$^{-3}$ (for $\beta_0 = 0$). It is thus evident that Chandrasekhar's general relativistic critical mass of $1.42$ M$_\odot$ [50] remains practically unaffected. Moreover, since this critical density $\rho_c^*$ is much below nuclear matter density, approximately $10^{14}$ g cm$^{-3}$, our consideration of a free fermionic equation of state remains valid throughout the stable regime of white dwarfs.

In the context of the stability analysis, we can analyse the mass–radius curve obtained in §3. Since the Chandrasekhar limit exists only below $\bar{\beta}_0 = 5.38 \times 10^{39}$, all mass–radius plots in figure 1 above this value would not correspond to reality. This is also evident from the fact that $\bar{\beta}_0$ is much higher than the electroweak bound $\beta_0 \sim 10^{34}$. For $\beta_0 < \bar{\beta}_0$, the mass–radius curves develop maximal points (figure 2a) at which the eigenfrequencies $\omega_0^2$ vanish as shown later in §4. These maximal points correspond to limiting Chandrasekhar mass lying below $\sim 1.425$ M$_\odot$. It is important to note that *the radius decreases as the mass increases* in the part of a mass–radius curve towards the right of the maximal point that corresponds to the *stable* branch. The mass–radius behaviour in the stable branches is consistent with several astronomical observations of white dwarfs [13–15,54–58]. Moreover, our stability analysis suggests that upon reaching beyond the Chandrasekhar mass the star would collapse to form highly dense compact objects such as a neutron star or black hole.

The present scenario of describing white dwarfs in terms of general relativity and GUP-enhanced equation of state with a positive GUP parameter $\beta$ rigorously leads to the existence of Chandrasekhar mass as well as the correct behaviour of the mass–radius relation consistent with astronomical observations. Moreover, it suggests the onset of gravitational collapse beyond the Chandrasekhar mass. It is now well-known that the degenerate core of a Type II supernova progenitor undergoes a gravitational collapse with a mass of about $1.4$ M$_\odot$, leading to the formation of a neutron star or a black hole.

Data accessibility. Numerical code and data to replicate the findings of this study are available within Dryad (https://doi.org/10.5061/dryad.dncjsxkzt) and published at Zenodo (https://doi.org/10.5281/zenodo.4625488) DOI: 10.5061/dryad.dncjsxkzt [59].

Authors' contributions. A.M. carried out this work with the supervision of M.K.N.
Competing interests. The authors declare that there are no competing interests.
Funding. This research article is supported by funding from the Indian Institute of Technology Guwahati, India.
Acknowledgements. A.M. is indebted to the Indian Institute of Technology Guwahati for extending various facilities of the Institute during his doctoral programme. M.K.N. thankfully acknowledges financial support from the Indian Institute of Technology Guwahati.

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
