## [Peer Review File · Royal Society Open Science]

Review History

RSOS-210301.R0 (Original submission)

Review form: Reviewer 1

Is the manuscript scientifically sound in its present form?

Yes

Are the interpretations and conclusions justified by the results?

Yes

Is the language acceptable?

Yes

Do you have any ethical concerns with this paper?

No

Have you any concerns about statistical analyses in this paper?

No

Recommendation?

Accept as is

Comments to the Author(s)

The authors have improved on the manuscript sufficiently enough in my opinion, and it can now be published.

Review form: Reviewer 2

Is the manuscript scientifically sound in its present form?

Yes

Are the interpretations and conclusions justified by the results?

Yes

Is the language acceptable?

Yes

Do you have any ethical concerns with this paper?

No

Have you any concerns about statistical analyses in this paper?

No

Recommendation?

Accept as is

Comments to the Author(s)

The authors have improved their manuscript, and hence I think that the present version is suitable for publication in Royal Society Open Science.

Decision letter (RSOS-210301.R0)

Dear Dr Nandy:

I am pleased to inform you that your manuscript entitled "Existence of Chandrasekhar's limit in generalised uncertainty white dwarfs" is now accepted for publication in Royal Society Open Science.

on behalf of Professor Stephen Smartt (Associate Editor) and Dr Rob Ivison (Subject Editor).

Reviewer(s)' Comments to Author:

Reviewer: 1

Comments to the Author(s)

The authors have improved on the manuscript sufficiently enough in my opinion, and it can now be published.

Reviewer: 2

Comments to the Author(s)

The authors have improved their manuscript, and hence I think that the present version is suitable for publication in Royal Society Open Science.
